# Collagen Mimetic Peptides Promote Adherence and Migration of ARPE-19 Cells While Reducing Inflammatory and Oxidative Stress

**DOI:** 10.3390/ijms23137004

**Published:** 2022-06-23

**Authors:** Marcio Ribeiro, Silvia Pasini, Robert O. Baratta, Brian J. Del Buono, Eric Schlumpf, David J. Calkins

**Affiliations:** 1Department of Ophthalmology and Visual Sciences, Vanderbilt Eye Institute, Vanderbilt University Medical Center, AA7103 MCN, 1161 21st Ave. S., Nashville, TN 37232, USA; marcio.ribeiro@vumc.org (M.R.); silvia.pasini@vumc.org (S.P.); 2Stuart Therapeutics, Inc., 411 SE Osceola St., Suite 203, Stuart, FL 34994, USA; bob@stuarttherapeutics.com (R.O.B.); brian@stuarttherapeutics.com (B.J.D.B.); eric@stuarttherapeutics.com (E.S.)

**Keywords:** ocular collagen, collagen mimetic peptides, retinal pigment epithelium, macular degeneration, extracellular matrix, ocular inflammation, Bruch’s membrane, cytokines, oxidative stress

## Abstract

Epithelial cells of multiple types produce and interact with the extracellular matrix to maintain structural integrity and promote healthy function within diverse endogenous tissues. Collagen is a critical component of the matrix, and challenges to collagen’s stability in aging, disease, and injury influence survival of adherent epithelial cells. The retinal pigment epithelium (RPE) is important for maintaining proper function of the light-sensitive photoreceptors in the neural retina, in part through synergy with the collagen-rich Bruch’s membrane that promotes RPE adherence. Degradation of Bruch’s is associated with RPE degeneration, which is implicated early in age-related macular degeneration, a leading cause of irreversible vision loss worldwide. Collagen mimetic peptides (CMPs) effectively repair damage to collagen helices, which are present in all collagens. Our previous work indicates that in doing so, CMPs promote survival and integrity of affected cells and tissues in models of ocular injury and disease, including wounding of corneal epithelial cells. Here, we show that CMPs increase adherence and migration of the ARPE-19 line of human RPE cells challenged by digestion of their collagen substrate. Application of CMPs also reduced both ARPE-19 secretion of pro-inflammatory cytokines (interleukins 6 and 8) and production of reactive oxygen species. Taken together, these results suggest that repairing collagen damaged by aging or other pathogenic processes in the posterior eye could improve RPE adherence and survival and, in doing so, reduce the inflammatory and oxidative stress that perpetuates the cycle of destruction at the root of age-related diseases of the outer retina.

## 1. Introduction

The extracellular matrix (ECM) is important as a physiologically active substrate for maintaining tissue stability and availability of signals between cells and the tissues they support [1]. Collagen is an important component of ECM for most tissues, and both injury and chronic disease influence the stability, turnover, and composition of collagen in affected structures [2]. Damage to collagen contributes to pathogenesis in many conditions affecting the eye and proximal structures in the optic projection to the brain [1]. In the eye, collagen distributes broadly, spanning structures in the anterior cornea to the optic nerve head in the posterior segment. In the posterior eye, the ECM of the retina and optic nerve contain concentrations of collagen types I, III, IV, V, and VI [3,4].

The retinal pigment epithelium (RPE) is a critical physiological partner, with multiple components of the outer retina. These include both neural elements anterior to the RPE (i.e., the photoreceptors) and vascular elements posterior (i.e., the choroid). The RPE is a monolayer of hexagonally arranged pigmented cells that performs a multitude of key functions necessary for maintaining normal vision, including renewal of light-sensitive photopigments whose conformational changes underlie the capacity of photoreceptors to signal contrast. The RPE also supplies biomolecules, nutrients, and oxygen to photoreceptors through the collagen-rich Bruch’s membrane, which serves as its ECM [5]. Bruch’s membrane is composed of two layers of collagen (fibrillar types I and III) that sandwich a middle layer composed primarily of elastin. The inner collagenous layer is adjacent to the basement membrane of the RPE, while the outer collagenous layer is associated with the basement membrane of the endothelial cells of the choriocapillaris [6]. Photoreceptor oxidative activity drives demand for degradative activity from RPE cells, the failure of which causes accumulation of drusen and other debris within the RPE layer as well as within the inner collagenous layer of Bruch’s membrane [7]. Damage to Bruch’s membrane is implicated in inflammation, complement activation, and oxidative stress in the RPE/choriocapillaris complex [8]. Over time, rejuvenation or repair of Bruch’s membrane could be a target to improve photoreceptor health and preservation of macular structure and function [6]. This highlights the importance of Bruch’s membrane in promoting the survival and function of RPE cells, whether endogenous or transplanted as part of experimental therapies for retinal disease, including age-related macular degeneration [5]. In turn, RPE cells are critical for producing and maintaining a healthy Bruch’s membrane, the inner-most layer of which comprises the RPE basement membrane, necessary for adherence [9,10]. The synergy between the RPE and Bruch’s membrane demonstrates the strong interdependence between the two adjacent tissues, one cellular and one acellular, highly concentrated in collagen [11].

A characteristic structural feature of all collagens is the common presence of their triple helical geometry, in which a set of three polypeptide chains comprising repeating sequences, typically of glycine, proline, and hydroxyproline triplets [12]. Previously, we and others have demonstrated the broad capacity for collagen mimetic peptides (CMPs) to repair damaged helical collagen of ECM in various therapeutic contexts, especially those intercalating sequences of type I collagen, with its high bioavailability, low antigenicity, and adaptability for various therapeutic platforms [13,14]. We have focused on the applicability and efficacy of CMPs in experimental models of ocular disease and injury, utilizing single strand CMPs specifically designed to reform native triple helices that have been damaged or partially digested by matrix metalloproteinases (MMPs) [15,16,17]. This is important work not only from the standpoint of restoring ECM structure, but also for reducing inflammation in local tissues [1]. We found that a CMP was efficacious in promoting the repair and structural integrity of the corneal epithelium following an acute injury [18]. Additional members of this CMP family promoted neurite outgrowth from dorsal root ganglia cultures following MMP-induced damage to the underlying collagen substrate [19].

Here, we tested the capacity of members of our CMP library to promote adherence and migration of the ARPE-19 line of human RPE cells. This cell line forms epithelial monolayers under growth-promoting conditions, demonstrates gene expression profiles similar to endogenous RPE, and is widely used in mechanistic studies of RPE function, motility, and pathogenesis [20,21,22,23]. In aging or disease, RPE cells struggle to maintain attachment to the collagenous layer of Bruch’s membrane, which leads to diminished capacity to maintain photoreceptor function and eventual RPE death [24,25,26]. Similarly, experimental therapies that attempt to replace lost RPE cells rely heavily on the capacity of Bruch’s membrane to promote adherence and migration [25,27]. We found that CMPs promote adherence and migration of ARPE-19 cells challenged by MMP digestion of the collagen substrate and provide an efficacious means to reduce secretion of pro-inflammatory mediators and oxidative stress.

## 2. Results

### 2.1. Collagen Mimetic Peptides Promote ARPE-19 Cell Adherence and Migration

Phase contrast micrographs of ARPE-19 cells on intact collagen showed a high level of adherence compared to those maintained on degraded collagen, which showed large patches devoid of cells (Figure 1A,B). In contrast, adherence of ARPE-19 cells on MMP-1-damaged collagen subsequently treated with two distinct CMPs (CMP 05A and CMP 13A, each at 100 µM) appeared to improve (Figure 1C,D).

We quantified ARPE-19 adherence relative to MMP-1 digested collagen for four distinct CMPs (Figure 2). Compared to intact collagen, MMP-1 digestion reduced adherence by 58% (*p* < 0.001). Treatment with both CMP 13A and CMP 09C increased adherence significantly, nearly two-fold for CMP 13A and by 37% for CMP 09C (*p* ≤ 0.04). CMPs 05A and 10A did not have a significant effect (*p* = 0.09). Following treatment, adherence with CMP 13A was similar to that of intact collagen (*p* = 018).

Recently, we found that a similarly structured CMP promoted repair of the corneal surface by enhancing migration of epithelial cells in vivo [18]. Here, we tested whether this same CMP(CMP 03A) similarly promotes adherence and migration of ARPE-19 cells using a transwell assay consisting of membrane inserts in which we coated only the basal (distal) side. Following application of ARPE-19 cells to the apical (proximal) side of the insert, coating with CMP 03A appeared to greatly enhance migration and adherence to the basal side compared to both vehicle and intact collagen itself (Figure 3).

We quantified adherence of Cresyl violet-stained cells following migration to the coated basal side of the insert cells applied to the apical side (Figure 4). Compared to vehicle, coating with collagen improved migration by almost 20% (*p* = 0.032). Coating with CMP 03A increased migration by 45% compared to vehicle and by 25% compared to collagen—both highly significant improvements (*p* < 0.001).

### 2.2. Collagen Mimetic Peptides Modulate ARPE-19 Cell Inflammatory and Oxidative Stress Mediators

Next, we tested whether CMP application could modulate secretion of the inflammatory mediators, interleukins 6 and 8 (IL-6 and IL-8, respectively), both of which are released by ARPE-19 cells in vitro and are relevant for RPE pathology under disease-relevant conditions (Holtkamp et al., 1998; Juel et al., 2012). Measurements using ELISA show that ARPE-19 cells maintained for 6 h on collagen degraded by MMP-1 (60 nM) increased IL-6 production by nearly five-fold compared to intact collagen (Figure 5A; 123.9 ± 14.9 vs. 16.9 ± 14.7 pg/mL, *p* < 0.001). Following MMP-1 exposure, subsequent application of 200 µM CMP 13A reduced IL-6 levels to control collagen (*p* = 0.99). We then exposed ARPE-19 cells to a lesser concentration of MMP-1 (20 nM) for 24 h, which also elicited a significant elevation in Il-6 compared to intact collagen (Figure 5B; 140.4 ± 3.8 vs. 39.2 ± 2.9 pg/mL, *p* < 0.001). In this case, concurrent application of CMP with MMP-1 was less effective at reducing IL-6. Even so, CMP 13A, either alone or in concert with CMP 03A, diminished IL-6 significantly compared to MMP-1 (*p* ≤ 0.014). Adding CMP 13A at the 12 h mark for the remaining 24 h period also reduced IL-6 (*p* = 0.001). Following 6 h treatment with MMP-1, IL-8 secretion from ARPE-19 cells was less than Il-6, but nevertheless significant compared to intact collagen (Figure 5C; 96.8 ± 4.2 vs. 65.6 ± 4.9 pg.ml, *p* < 0.001). Once again, subsequent treatment with CMP 13A significantly reduced IL-8—in this case, to about 20% of intact collagen (13.4 ± 3.1 pg/mL, p < 0.001). Interestingly, over 24 h, MMP-1 had little influence on IL-8 secretion compared to intact collagen, and concurrent application of CMP did not modulate levels (Figure 5D, *p* = 0.30). For both IL-6 and IL-8, application of lipopolysaccharide (10 mg/mL) as a positive control elevated secretion far above MMP-1 levels.

Fragmentation of collagen induces oxidative stress in a variety of tissues, which, in turn, can further activate MMPs [28,29]. Interestingly, even though oxidative stress is a major component of RPE senescence and degeneration [10,30], ARPE-19 cells appear to be relatively resistant to this stress and do not undergo caspase 3-dependent apoptosis as a result [23,31]. We used a fluorescent indicator of reactive oxygen species (ROS) formation (carboxy-2′,7′-dichlorodihydrofluorescein diacetate, or DCF) to test whether CMP application influences the oxidative response of ARPE-19 cells. Application of MMP-1 (120 nM) to type I collagen did not significantly change levels of DCF fluorescence compared to intact collagen (Figure 6, *p* = 0.57), consistent with ARPE-19 cells’ low susceptibility to oxidative stress [23,31]. Accordingly, under the conditions tested, caspase-3 activity did not change even with MMP-1 application (Figure 6, right). Interestingly, for both intact collagen and MMP-1 treated collagen, CMP 13A significantly lowered the baseline DCF signal (*p* ≤ 0.04), suggestive of an antioxidative influence.

## 3. Discussion

Accumulating evidence indicates that repairing the helical organization of fragmented collagen strands could restore both the biomechanical and structural roles of extracellular membranes and matrix and, in doing so, restore homeostatic anti-inflammatory signaling [1]. Our work with the mimetics of helical collagen has demonstrated broad capacity for their use in tissue repair in the eye and optic projection to the brain. In acute corneal wounds, topical application of CMP (CMP 03A) accelerated regeneration and migration of the epithelial layer in part by enhancing the adherence of migrating cells to the underlying stroma [18]. We also discovered that multiple CMPs (including CMP 13A and CMP 09C, used here) promoted neurite outgrowth from dorsal root ganglia plated on collagen partially digested by MMP-1 [19]. Finally, intravitreal delivery of CMPs rescued axonal transport along the optic nerve in experimental glaucoma and promoted recovery of axons damaged by acute injury to the nerve [19,32].

Our key finding here is that CMPs may have utility in repairing damaged collagen in the outer retina, as well. Under conditions of collagen challenged by MMP digestion, multiple CMPs (CMP 13A and CMP 09C) promoted adherence of ARPE-19 cells maintained in culture (Figure 1 and Figure 2). Similarly, another CMP (CMP 03A) promoted the migration of ARPE-19 cells across a transwell membrane (Figure 3 and Figure 4), which is reminiscent of this same mimetic’s positive influence on corneal epithelial migration and adherence [18]. The application of CMP 13A also reduced secretion of inflammatory mediators (IL-6 and IL-8) from ARPE-19 cells challenged by MMP-1, as measured by ELISA (Figure 5), though the influence on IL-8 levels was insignificant with concurrent exposure to MMP-1 (Figure 5D). Finally, CMP 13A also reduced baseline ROS formation in ARPE-19 cells (Figure 6).

These results could have important implications for the therapeutic applications of collagen mimetics in diseases of the posterior eye, especially age-related macular degeneration, the atrophic form of which involves early degeneration of the RPE with subsequent loss of photoreceptors in the neural retina [10]. Susceptibility is augmented by accumulation of age-related changes in the integrity of Bruch’s membrane, which provides structural and functional support to the RPE [33,34]. Bruch’s membrane is a highly organized collagen-rich basement membrane that maintains separation of RPE from the underlying choroidal capillaries [6]. Restoration of Bruch’s membrane is an integral part of efforts to promote adherence and migration of transplanted RPE cells as a potential therapy for macular degeneration [35], while RPE cells synthesize the most abundant proteins of Bruch’s membrane, especially type I collagen and laminin, as well as the MMPs and their inhibitors involved in ECM remodeling [9]. This symbiotic relationship highlights the importance of repairing endogenous collagen as an early step in restoring RPE integrity.

The RPE is a critical component of the blood–retina barrier and, in this capacity, is important in mediating retinal immune responses. RPE cells produce pro-inflammatory cytokines in polarized fashion, directed towards the choroid as a possible means of protecting the privilege of the neural retina from infiltrating immune cells [36]. In response to stress, RPE cells secrete several cytokines, including interleukins 6 and 8 (IL-6, IL-8) [37,38], and RPE-mediated inflammation is thought to be pathogenic in macular degeneration [8,38]. Conversely, exposure to tumor necrosis factor α (TNFα) causes apoptosis of ARPE-19 cells along with secretion of multiple inflammatory cytokines; both can be reversed with treatment using a potent inhibitory of the transcription factor NF-κB [38]. Secretion of VEGFA (vascular endothelial growth factor A) from RPE cells is pathogenic in macular degeneration and arises in part through the cytokine Transforming Growth Factor β (TGFβ), which also changes the migratory and adherent capacity of RPE cells [39,40]. Treatment with CMP 13A subsequent to MMP-1 induced elevations in IL-6 and IL-8 significantly reduced secretion of both (Figure 5), suggesting that collagen repair could quench other pro-inflammatory processes in the RPE layer that are implicated in degeneration. Along with inflammation, oxidative stress is a known pathogenic contributor to RPE degeneration in aging, which increases susceptibility to macular degeneration [10]. That CMP reduces ROS formation in ARPE-19 maintained in vitro (Figure 6) suggests that remodeling collagen in Bruch’s membrane could slow the cycle of MMP activation and oxidative stress that results from fragmentation [28,29]. Taken together, our results suggest that a direct collagen reparative, which does not exist in natural disease repair, could provide a rapid and effective enhancement of recovery of the synergistic relationship between RPE cells and Bruch’s membrane.

## 4. Materials and Methods

### 4.1. ARPE-19 Cell Cultures

No animals were used in these studies. We maintained ARPE-19 cells (ATCC CRL2302, Manassas, VA, USA) in DMEM/F-12, GlutaMAX™ medium supplement with 10% FBS (Thermo Scientific, Waltham, MA, USA) and 1% penicillin or streptomycin and incubated at 37 °C in a humidified atmosphere with 5% CO_2_. The cells were used for experiments after six passages at a density of 150,000 cells/mL. To prepare partially digested collagen, we activated 60 nM human metalloproteinase 1 (MMP-1, Biolegend, San Diego, CA, USA) with 2 mM 4-aminophenylmercuric acetate (APMA, Sigma, St. Louis, MO, USA) for 30 min at 37 °C in TCN buffer (25 mM Tris-HCl, 10 mM CaCl_2_, 150 mM NaCl, pH 7.5), following an established protocol used in our previous study [19,41]. We added human type I atelocollagen (3.375 mg, 0.2 mg/mL, Advance BioMatrix, Carlsbad, CA, USA) and incubated the mixture at 37 °C for an additional 6 h. We confirmed collagen digestion with SDS-PAGE using 4–20% polyacrylamide gels (Bio-Rad Laboratories, Hercules, CA, USA) under reducing conditions, as shown previously [19]. We utilized four distinct single-strand, 21-residue collagen mimetic peptides (CMPs), analogous to CMPs, that intercalate with high affinity and selectivity into damaged type I collagen in vitro and in vivo [15,16,17]; these were the same as those manufactured in limited quantity by Bachem, AG (Torrance, CA, USA), used in our previous studies of dorsal root ganglia cultures [19]. These were: CMP 03A (Pro-Pro-Gly)_7_, CMP 05A (Pro-Hyp-Gly)_7_, CMP 09C (Pro-trans-Flp-Gly)_6_-Pro-Cys-Gly, CMP 10A (Hyp-Pro-Gly)_7_-P, and CMP 13A (cis-Flp-Hyp-Gly)_7_, where abbreviations indicate proline (Pro), hydroxyproline (Hyp), glycine (Gly), 4-fluoro-proline (Flp), and the undecapeptide tachykinin neuropeptide substance P (P).

### 4.2. Adherence and Migration Assays

To measure the influence of CMP application on ARPE-19 cellular adherence, we diluted the mixture of activated MMP-1 and type 1 atelocollagen, described above, or pure type I atelocollagen with double-distilled water (DDW) to bring the concentration of total collagen to 100 µg/mL. We coated 48-well plates using 100 µL of collagen solution and 60,000 ARPE-19 cells per well and incubated the plates at 37 °C overnight in serum free media (12 h). The following day, we aspirated the collagen solution and washed the plates three times with DDW. We applied either CMP (100 µM, 100 µL) or vehicle (DDW) and incubated once again at 37 °C for 5 h before removing the solutions and washing again three times with DDW. We detached the adherent cells with 200 µL Tryple Express (Thermo Scientific, Waltham, MA) and counted them with the hemacytometer.

We also tested how CMP application affected ARPE-19 cell adherence and migration compared to intact collagen using a transwell migration assay (Costar transwell inserts; 8 mm pore size, 6.5-mm membrane, MilliporeSigma, Burlington, MA, USA), which has been used previously for ARPE-19 cells in other contexts (Holtkamp et al., 1998). The basal (distal) side of the inserts was coated for 24 h in 500 µL of substrate, either vehicle (DDW), human type 1 atelocollagen (100 µg/mL), or CMP 03A (100 µM), which was chosen because of our previous studies demonstrating its efficacy in healing corneal epithelial cells [18]. The inserts were washed with phosphate-buffered saline (PBS) and allowed to dry at 37 °C. We resuspended 1 × 10^5^ ARPE-19 cells in 100 µL serum-free media and applied them only to the apical (proximal) side of each precoated insert membrane. We then placed the inserts in a 24-well plate containing 0.5 mL per well of complete media plus 10% FBS, to attract cells and promote migration across the insert membrane, and incubated for 4 h at 37 °C. Following incubation, non-migrated cells on the apical side of the insert were carefully removed with a cotton swab. Cells that successfully migrated and adhered to the basal side of the membrane were fixed with 4% paraformaldehyde for 5 min at room temperature, stained with 0.1% Cresyl violet solution for 5 min, and then washed gently with water. The number of cells migrating through and adhering to the basal membrane was determined by counting cells using a custom MATLAB program within multiple 10× and 20× magnification fields using phase-contrast microscopy.

### 4.3. ELISA

We reconstituted 0.1 mg/mL of human type I collagen solution (VitroCol^®^, Advanced BioMatrix, Carlsbad, CA, USA) in 0.01 M HCl and sterile 0.25% acetic acid, which was added to wells (24-well plates) and incubated for 2 h at room temperature. After incubation, coated surfaces were washed twice with PBS before seeding at a density of 3 × 10^5^ ARPE-19 cells/mL in serum media, as described above. After 24 h, the media were replaced with serum-free media. We added to the cells MMP-1 (60 or 20 nM), activated as described above, and/or 200 µM of CMP (in DDW, vehicle), following different time courses (see Results). Upon completion of the experiments, supernatants were collected and used to measure IL-6 (Human IL-6 ELISA Kit, Abcam, Waltham, MA, USA) and IL-8 (Human IL-8 ProQuantum, Thermo Fisher, Nashville, TN, USA), following the manufacturer’s instructions.

### 4.4. Reactive Oxygen Species and Caspase-3 Measurements

We prepared partially digested human type I collagen as described above, using activated MMP-1 (120 nM), and plated ARPE-19 cells, as described, using 96-well plates. We applied 10 µM 6-carboxy-2′,7′-dichlorodihydrofluorescein diacetate (abbreviated as DCF) as a general oxidative stress indicator and marker for reactive oxygen species (# C400, Thermo Scientific) to the live cells for 30 min at 37 °C, following published protocols [42,43]. After incubation, the media were washed and relative fluorescence measured (excitation/emission: 493/520 nm) using a multi-well plate reader (time 0). The cells were maintained at 37 °C for an additional 25 min and relative fluorescence measured again (time 25). The results were expressed as the difference between DCF fluorescence at time 25 and time 0, which was normalized to the total protein amount in µg per sample.

Caspase-3 activity as an index of apoptosis was measured using the EnzChek™ Caspase-3 Assay Kit #1, Z-DEVD-AMC substrate (#E13183, Thermo Fisher, Nashville, TN, USA). Briefly, cells treated as described in the preceding paragraph were lysed using 50 µL of 1× Cell Lysis Buffer (Component C from the Kit). The lysates were then frozen at –80 °C and thawed in ice before adding 50 µL of the 2× substrate working solution (0.2 mM Z-DEVD-AMC substrate (Component A from the kit), 2× reaction buffer (Component D), and 10 µM DTT (Component E) to each sample. The samples were incubated for 30 min at room temperature and the fluorescence was measured (excitation/emission: 342/441 nm) using a multi-well plate reader. The samples (1 µL of sample diluted in 24 µL H_2_O) were then used to quantify the total protein levels using the Pierce™ BCA Protein Assay Kit (# 23225 Thermo Fisher, Nashville, TN, USA), and the caspase-3 activity values (relative fluorescence units, RFU) were normalized to the µg of protein for each sample.

### 4.5. Statistical Analysis

All data are presented as mean ± standard error of the mean (SEM). Statistical analyses and graphs were made using Sigma Plot Version 14 (Systat, San Jose, CA, USA). Outlier analysis was performed using Grubbs’ test (Graphpad Software, San Diego CA, USA). Parametric statistics were performed (*t*-test, analysis of variance) if data passed normality and equal variance tests; otherwise, we performed non-parametric statistics (Mann–Whitney, ANOVA on Ranks, Welch’s test). For all experiments described, measurements were made from no less than six independent samples, all analyzed blindly. Statistical significance was defined as *p* ≤ 0.05.

## Figures and Tables

**Figure 1 ijms-23-07004-f001:**
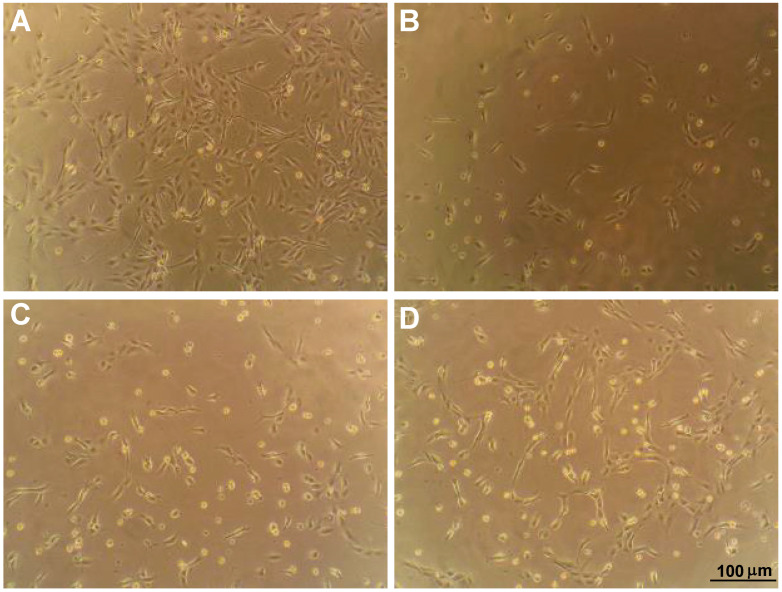
Mimetic peptides promote adherence of ARPE-19 cells. Phase-contrast microscopy of preparations of ARPE-19 cells maintained in vitro as described after plating on intact human type 1 atelocollagen (**A**) vs. collagen partially digested with activated MMP-1 (**B**). Treatment of cells plated on MMP-1 digested collagen with CMP 05A (**C**) and CMP 13A (**D**), both at a concentration of 100 µM. CMP 13A appears to improve adherence. Scale for each panel as indicated (**D**).

**Figure 2 ijms-23-07004-f002:**
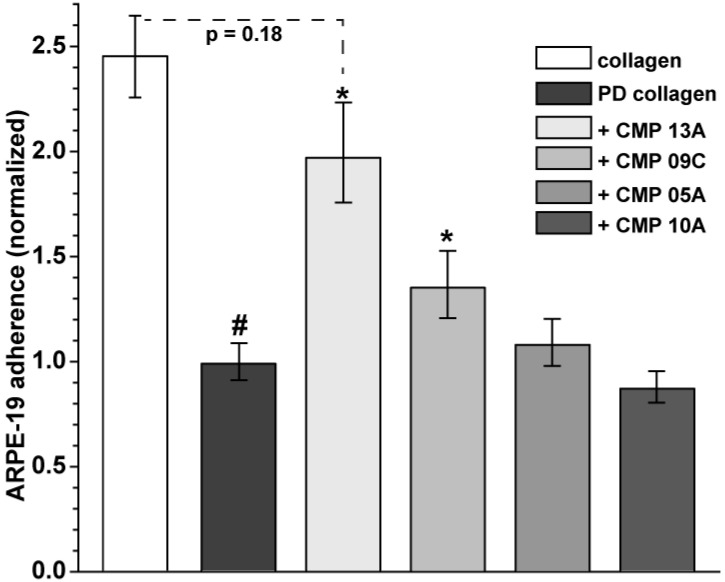
Quantification of ARPE-19 cell adherence. Adherence of ARPE-19 cells plated on collagen partially digested with MMP-1 as described and treated with 100 µM of CMP as indicated. MMP-1 digestion diminished adherence by 58% compared to intact collagen (#, *p* < 0.001). When normalized to untreated partially digested (PD) collagen, both CMP 13A and CMP 09C increased adherence significantly (*, *p* ≤ 0.04), and adherence with CMP 13A did not differ from intact collagen (*p* = 0.18). Data = mean ± SEM (*n* ≥ 61 samples for each condition).

**Figure 3 ijms-23-07004-f003:**
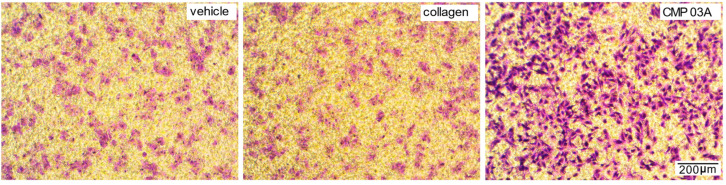
CMP promotes migration and adherence of ARPE-19 cells in transwell assay. Cresyl violet-stained ARPE-19 cells migrating and adhering to the pre-coated basal (distal) side of transwell membranes, following application of cells to the apical side only. Migration and adherence appeared significantly higher for basal surfaces coated with CMP 03A. Scale in lower right panel applies to all images.

**Figure 4 ijms-23-07004-f004:**
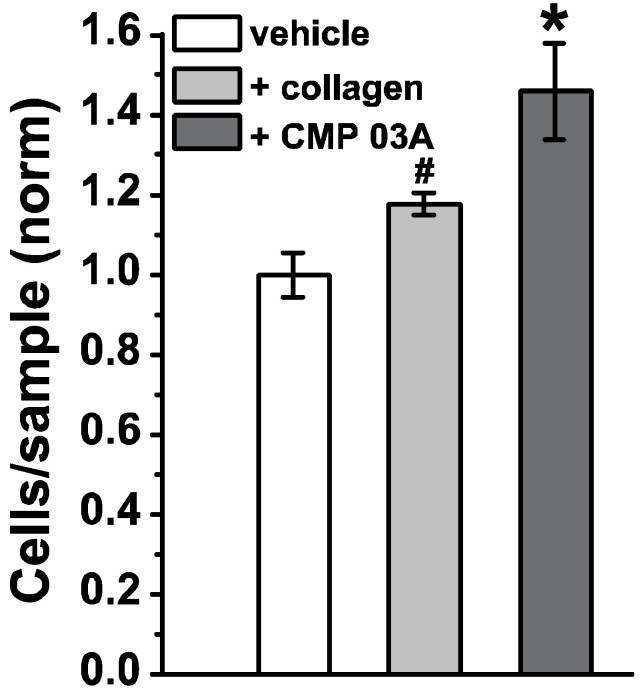
Quantification of ARPE-19 cell migration. ARPE-19 cells adhering to the basal side of transwell membranes which were pre-coated with either vehicle (DDW), intact human type I collagen, or CMP 03A (100 µM) following application of cells to the apical side. Normalized relative to vehicle, CMP 03A promoted migration and adherence significantly better than both vehicle and collagen (*, *p* < 0.001), while collagen was significantly more efficacious than vehicle (#, *p* = 032). Data = mean ± SEM (*n* ≥ 6 samples for each condition).

**Figure 5 ijms-23-07004-f005:**
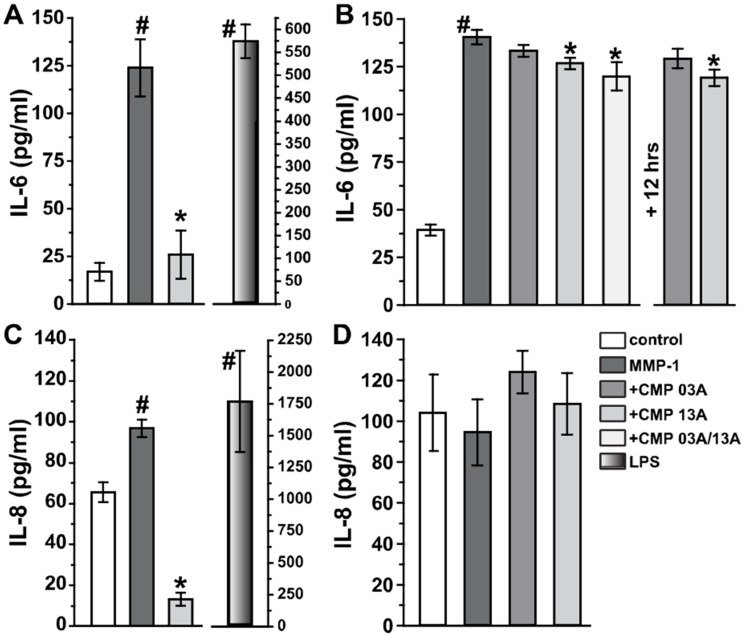
CMP treatment reduces ARPE-19 inflammatory signaling: ELISA measurements. (**A**) Interleukin-6 (IL-6) secretion from ARPE-19 cells increased significantly following treatment with activated MMP-1 (60 nM) for 6 h compared to intact collagen control (#, *p* < 0.001). Following a complete exchange of media, cells were treated with 200 µM CMP 13A, which reduced Il-6 by 5-fold (*, *p* = 0.007) to control levels (*p* = 0.99). Right: positive control using lipopolysaccharide (LPS, 10 mg/mL) shows +30-fold increase compared to control (*p* <0.001). (**B**) Left: IL-6 secretion increased significantly following 24 h of 20 nM MMP-1 (#, *p* < 0.001). Concurrent treatment with CMP 13A significantly reduced Il-6 compared to MMP-1 (*p* = 0.005), as did a combination of CMP 13A and 03A (*p* = 0.014, concentration of each was 200 µM). Right: adding 200 µM CMP 13A at the 12 h mark for the remaining 12 h also reduced Il-6 compared to MMP-1 (*p* = 0.001). (**C**) Interleukin-8 (IL-8) secretion from ARPE-19 cells also increased following MMP-1(6 h, 60 nM) compared to control (#, *p* < 0.001). Following complete media exchange, CMP 13A reduced Il-8 compared to both MMP-1 and control (*, *p* < 0.001). LPS positive control (right) induced a nearly 30-fold increase in IL-8 compared to control (#, *p* < 0.001). (**D**) IL-8 secretion did not change following 24 h of 20 nM MMP-1 alone or with concurrent treatment with CMP compared to control (*p* = 0.30). For each measurement, *n* ≥ 6.

**Figure 6 ijms-23-07004-f006:**
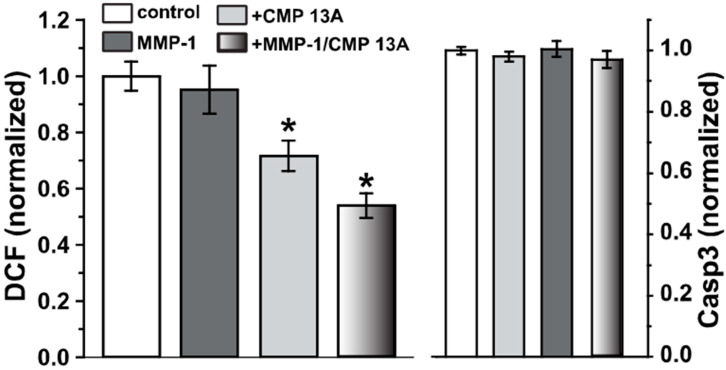
CMP treatment reduces ARPE-19 oxidative stress. Left: fluorescent signal from dichlorodihydrofluorescein (DCF) treated ARPE-19 cells as a measure of reactive oxygen species (ROS). Application of CMP 13A (200 µM) to either intact collagen or collagen digested by MMP-1 (120 nM) significantly reduced levels of ROS compared to both intact control and untreated MMP-1 digested collagen (*, *p* ≤ 0.04); the two CMP 13A conditions did not differ (*p* = 0.14). Right: no condition increased ARPE-19 apoptosis compared to control, as assessed by caspase-3 (Casp3) activity (*p* = 0.30). For each measurement, *n* ≥ 15.

## Data Availability

All the data reported in this study are shown in this manuscript.

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
