# Peer review of "Collagen Mimetic Peptides Promote Adherence and Migration of ARPE-19 Cells While Reducing Inflammatory and Oxidative Stress"

_ijms, 2022, doi:10.3390/ijms23137004_

Round 1

Reviewer 1 Report

Ms-ijms-1778437 can be accepted with minor revisions.

The introduction can be enhanced with information regarding the effects of TNFa and the TGFb family, factors that modulate the migratory activity of the cell lines treated by the authors, expanding the picture of active cytokines and not only IL6 and IL8 considered in the study. The images proposed in Fig. 1 in phase-contrast give a summary response, they could be accompanied by specific histological staining for collagen with aniline blue.

The graphs in Fig. 2 and 4 can be reduced and standardized

The conclusions drawn by the authors in the final discussion are adequate to the results obtained and presented.

The text must be double-checked for the presence of some typos and parts with different fonts.

The bibliographic sources include only four articles published in recent years.

Author Response

            Thank you kindly for your enthusiasm for our research article “Collagen Mimetic Peptides Promote Adherence and Migration of ARPE-19 Cells while Reducing Inflammatory and Oxidative Stress”.  With your thoughtful comments and edits, we have prepared a revision that we believe is now ready for publication in International Journal of Molecular Sciences for the special issue "The Collagen Connection".  Below is our point-by-point response.

Respectfully,

The Authors

Reviewer 1

Abstract: there are some grammatical errors, missing definite articles "The ..."

Thank you for pointing this out. We have fixed these occurrences.

The sentence:Our previous work indicates that collagen mimetic peptides (CMPs) effectively repair damage to collagen helices, now understood to be present in all collagens, to promote survival and integrity of affected cells and tissues in models of ocular injury and disease, including wounding of corneal epithelial cellsis difficult to read, please paraphrase.

We agree and have rewritten the passage as “Collagen mimetic peptides (CMPs) effectively repair damage to collagen helices, which are present in all collagens. Our previous work indicates that in doing so, CMPs promote survival and integrity of affected cells and tissues in models of ocular injury and disease, including wounding of corneal epithelial cells”.

The sentences on line 34 and line 36 start with the same phrase “Collagen is an important……” may need to be combined or paraphrased.

Thank you for pointing this out. We have reworded the second passage as “Damage to collagen contributes to pathogenesis in many conditions….”

Reviewer 2 Report

The manuscript titled: “Collagen Mimetic Peptides Promote Adherence and Migration of ARPE-19 Cells while Reducing Inflammatory and Oxidative Stressis very interesting and very new. I have only a few remarks:

Abstract: there are some grammatical errors, missing definite articles "The ..."

The sentence:Our previous work indicates that collagen mimetic peptides (CMPs) effectively repair damage to collagen helices, now understood to be present in all collagens, to promote survival and integrity of affected cells and tissues in models of ocular injury and disease, including wounding of corneal epithelial cellsis difficult to read, please paraphrase.

Introduction

The sentences on line 34 and line 36 start with the same phrase Collagen is an important……” may need to be combined or paraphrased.

Materials and Methods: are very well described

The results are clear and well organized

Author Response

Dear Colleagues,

            Thank you kindly for your enthusiasm for our research article “Collagen Mimetic Peptides Promote Adherence and Migration of ARPE-19 Cells while Reducing Inflammatory and Oxidative Stress”.  With your thoughtful comments and edits, we have prepared a revision that we believe is now ready for publication in International Journal of Molecular Sciences for the special issue "The Collagen Connection".  Below is our point-by-point response.

Respectfully,

The Authors

Reviewer 1

Abstract: there are some grammatical errors, missing definite articles "The ..."

Thank you for pointing this out. We have fixed these occurrences.

The sentence: “Our previous work indicates that collagen mimetic peptides (CMPs) effectively repair damage to collagen helices, now understood to be present in all collagens, to promote survival and integrity of affected cells and tissues in models of ocular injury and disease, including wounding of corneal epithelial cells” is difficult to read, please paraphrase.

We agree and have rewritten the passage as “Collagen mimetic peptides (CMPs) effectively repair damage to collagen helices, which are present in all collagens. Our previous work indicates that in doing so, CMPs promote survival and integrity of affected cells and tissues in models of ocular injury and disease, including wounding of corneal epithelial cells”.

The sentences on line 34 and line 36 start with the same phrase “Collagen is an important……” may need to be combined or paraphrased.

Thank you for pointing this out. We have reworded the second passage as “Damage to collagen contributes to pathogenesis in many conditions….”

Reviewer 2

The introduction can be enhanced with information regarding the effects of TNFa and the TGFb family, factors that modulate the migratory activity of the cell lines treated by the authors, expanding the picture of active cytokines and not only IL6 and IL8 considered in the study… The bibliographic sources include only four articles published in recent years.

            Thank you for pointing this out. We have added text in the discussion relevant to TNFa and TGFb and have added more recent references accordingly in the bibliography.

The graphs in Fig. 2 and 4 can be reduced and standardized.

            Thank you; we have done so.

The conclusions drawn by the authors in the final discussion are adequate to the results obtained and presented.

            Thank you kindly.

The text must be double-checked for the presence of some typos and parts with different fonts.

            Thank you for pointing these out; we have fixed several occurrences.
